# FUSION 360 GALLERY: A DATASET AND ENVIRONMENT FOR PROGRAMMATIC CAD RECONSTRUCTION

## ABSTRACT

Parametric computer-aided design (CAD) is a standard paradigm used for the design of manufactured objects. CAD designers perform modeling operations, such as *sketch* and *extrude*, to form a construction sequence that makes up a final design. Despite the pervasiveness of parametric CAD and growing interest from the research community, a dataset of human designed 3D CAD construction sequences has not been available to-date. In this paper we present the *Fusion 360 Gallery* reconstruction dataset and environment for learning CAD reconstruction. We provide a dataset of 8,625 designs, comprising sequential *sketch* and *extrude* modeling operations, together with a complementary environment called the *Fusion 360 Gym*, to assist with performing CAD reconstruction. We outline a standard CAD reconstruction task, together with evaluation metrics, and present results from a novel method using neurally guided search to recover a construction sequence from a target geometry.

## 1 INTRODUCTION

The manufactured objects that surround us in everyday life are created in computer-aided design (CAD) software using common modeling operations such as *sketch* and *extrude*. With just these two modeling operations, a highly expressive range of 3D designs can be created (Figure 1). Parametric CAD files contain construction sequence information that is critical for documenting design intent, maintaining editablity, and downstream simulation and manufacturing. Despite the value of this information, it is often lost due to data translation or error and must be reverse engineered from geometry or even raw 3D scan data. The task of reconstructing CAD operations from geometry has been pursued for over 40 years (Shah et al., 2001) and is available in commercial CAD software using heuristic approaches (Autodesk, 2012; Dassault, 2019). Recent advances in neural networks for 3D shape generation has spurred new interest in CAD reconstruction, due to the potential to generalize better and further automate this challenging problem. However, learning-based approaches

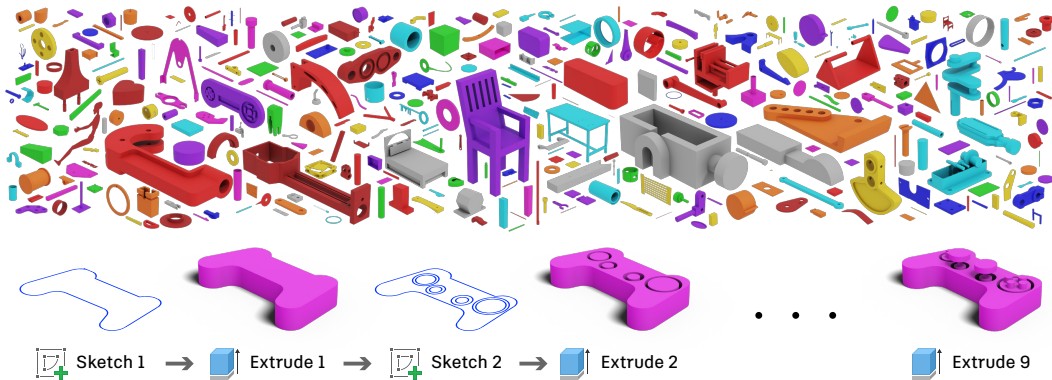

Figure 1: Top: A subset of designs containing 3D CAD construction sequences from the *Fusion 360 Gallery* reconstruction dataset. Bottom: An example construction sequence using *sketch* and *extrude* modeling operations.

to CAD reconstruction have not yet had access to a human-designed dataset of 3D CAD construction sequences, instead relying on synthetic data for both training and testing purposes, e.g. Li et al. (2020). The absence of real world data has limited work on CAD reconstruction using common sketch and extrude modeling operations. Instead a focus has been on reconstruction from simple geometric primitives (Sharma et al., 2017; Tian et al., 2019; Ellis et al., 2019) that lack the rich parametric sketches commonly used in mechanical CAD (e.g. Figure 2). As there is no existing learning-based approach to reconstruct sketch and extrude sequences, we take a first step towards this goal by introducing data, a supporting software environment, and a novel action representation for reconstructing sketch and extrude designs.

In this paper we present the *Fusion 360 Gallery* reconstruction dataset and environment for learning CAD reconstruction. The dataset contains 8,625 designs created by users of Autodesk Fusion 360 using a simple subset of CAD modeling operations: *sketch* and *extrude*. To the best of our knowledge this dataset is the first to provide human designed 3D CAD construction sequence data for use with machine learning. To support research with the dataset we provide an environment called the *Fusion 360 Gym* for working with CAD reconstruction. A key motivation of this work is to provide insights into the process of *how* people design objects. Furthermore, our goal is to provide a universal benchmark for research and evaluation of learning-based CAD reconstruction algorithms, bridging the gap between the computer graphics and machine learning community. To this end we describe a standard CAD reconstruction task and associated evaluation metrics with respect to the ground truth construction sequence. We also introduce a novel action representation for CAD reconstruction of sketch and extrude designs using neurally guided search. This search employs a policy, trained using imitation learning, consisting of a graph neural network encoding of CAD geometry.

This paper makes the following contributions:

- We present the *Fusion 360 Gallery* reconstruction dataset containing construction sequence information for 8,625 human-designed sketch and extrude CAD models.
- We introduce an environment called the *Fusion 360 Gym*, standardizing the CAD reconstruction task in a Markov Decision Process formulation.
- We introduce a novel action representation to enable neurally guided CAD reconstruction trained on real world construction sequences for the first time.

## 2 RELATED WORK

**CAD Datasets** Existing 3D CAD datasets have largely focused on providing mesh geometry (Chang et al., 2015; Wu et al., 2015; Zhou & Jacobson, 2016; Mo et al., 2019b; Kim et al., 2020). However, the de facto standard for parametric CAD is the boundary representation (B-Rep) format, containing valuable analytic representations of surfaces and curves suitable for high level control of 3D shapes. B-Reps are collections of trimmed parametric surfaces along with topological information which describes adjacency relationships and the ordering of elements such as faces, loops, edges, and vertices (Weiler, 1986). B-Rep datasets have recently been made available with both human-designed (Koch et al., 2019) and synthetic data (Zhang et al., 2018; Jayaraman et al., 2020; Starly, 2020). Missing from these datasets is construction sequence information. We believe it is critical to understand not only *what* is designed, but *how* that design came about.

Parametric CAD files contain valuable information on the construction history of a design. Schulz et al. (2014) provide a standard collection of human designs with full parametric history, albeit a limited set of 67 designs in a proprietary format. SketchGraphs (Seff et al., 2020) constrains the broad area of parametric CAD by focusing on the underlying 2D engineering sketches, including sketch construction sequences. In the absence of 3D human design data, learning-based approaches have instead leveraged synthetic CAD construction sequences (Sharma et al., 2017; Li et al., 2020). The dataset presented in this paper is, to the best of our knowledge, the first to provide human-designed 3D CAD construction sequence information suitable for use with machine learning.

**CAD Reconstruction** The task of CAD reconstruction involves recovering the sequence of modeling operations used to construct a CAD model from geometry input, such as B-reps, triangle meshes, or point clouds. Despite extensive prior work (Shah et al., 2001), CAD reconstruction remains a challenging problem as it requires deductions on both continuous parameters (e.g., ex-

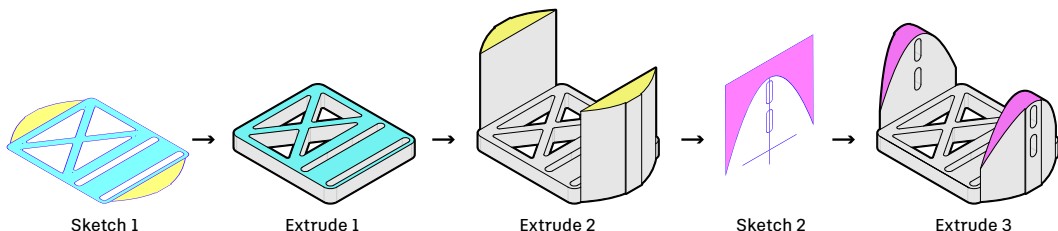

Figure 2: An example design sequence from the *Fusion 360 Gallery* reconstruction dataset. Sketch profiles are sequentially extruded to *join* (Extrude 1, Extrude 2) or *cut* (Extrude 3) geometry using Boolean operations. The colored areas show the sketch profiles that partake in each extrusion.

tracting the dimensions of primitives) and discrete operations (e.g., choosing a proper operation for the next step), leading to a mixed combinatorial search space. To recover the sequence of operations, traditional methods typically run global search methods (e.g., evolutionary algorithms as in Hamza & Saitou (2004), Weiss (2009), Friedrich et al. (2019), and Fayolle & Pasko (2016)) with heuristic rules to prune the search space (Shapiro & Vossler, 1993; Buchele, 2000; Buchele & Roles, 2001; Buchele & Crawford, 2003). Heuristic approaches are also available in a number of commercial software tools, often as a user-guided semi-automatic system (Autodesk, 2012; Dassault, 2019) to aid with file conversion between CAD systems. These traditional algorithms operate by removing faces from the B-rep body and reapplying them as parametric modeling operations. This strategy can recover the later modeling operations, but fail to completely rebuild the construction sequence from the first step. We instead tackle the task of recovering the entire construction sequence from the first extrusion. Another approach is using program synthesis (Du et al., 2018; Nandi et al., 2017; 2018; 2020) to infer CAD programs written in domain specific languages from given shapes. CAD reconstruction is also related to the inverse procedural modeling problem (Talton et al., 2011; Stava et al., 2014; Vanegas et al., 2012), which attempts to reverse-engineer procedures that can faithfully match a given target. Inverse procedural modeling methods typically work with synthetic data, while our paper tackles tasks on real CAD models and modeling operations.

Compared to the rule-based or grammar-based methods above, learning-based approaches can potentially learn the types of rules that are typically hard-coded, automate scenarios that require user-input, and generalize when confronted with unfamiliar geometry. One of the earliest such works is CSGNet (Sharma et al., 2017), which trains a neural network to infer the sequence of Constructive Solid Geometry (CSG) operations based on visual inputs. More recent works along this line of research include Ellis et al. (2019), Tian et al. (2019), and Kania et al. (2020). Typically associated with these methods are a customized, domain specific language (e.g., CSG) that parameterizes the space of geometry, some heuristic rules that limit the search space, and a neural network generative model (Zou et al., 2017; Mo et al., 2019a; Chen et al., 2020; Jones et al., 2020). Lin et al. (2020) reconstruct input shapes with a dual action representation that first positions cuboids and then edits edge-loops for refinement. Although editing edge-loops of cuboids may be a suitable modeling operation in artistic design, it is not as expressive or precise as the sketch and extrude operations used in real mechanical components. Additionally, Lin et al. (2020) chooses to train and evaluate their network on synthetic data due to the lack of a benchmark dataset of CAD construction sequences, a space that our work aims to fill. Our approach is, to the best of our knowledge, the first to apply a learning-based method to reconstruction using common sketch and extrude CAD modeling operations from real human designs.

## 3 FUSION 360 GALLERY RECONSTRUCTION DATASET

The *Fusion 360 Gallery* reconstruction dataset[1] is produced from designs submitted by users of the CAD software Autodesk Fusion 360. The dataset contains CAD construction sequence information from a subset of sketch and extrude designs. We intentionally limit the data to the *sketch* and *extrude* modeling operations to reduce the complexity of the CAD reconstruction task. Figure 1 shows a random sampling of the designs in the dataset. Each design is provided in three different representations: B-Rep, mesh, and construction sequence JSON text format. An official 80:20 train-

---

[1]https://github.com/xxx/xxx

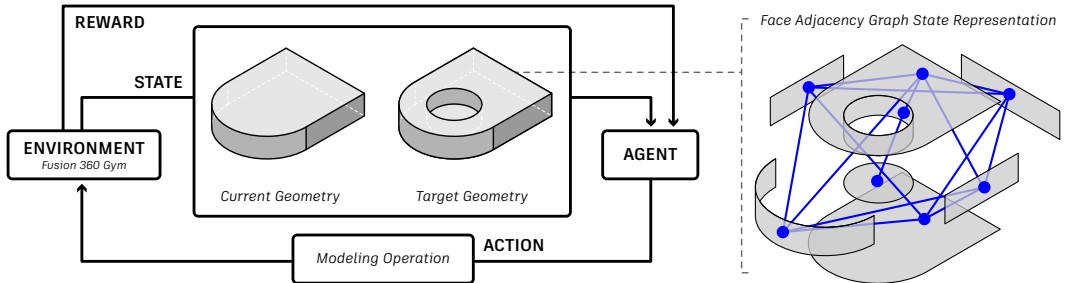

Figure 3: The *Fusion 360 Gym* interacts with an *agent* in a sequential decision making scenario (left) with the *state* containing geometries represented as face adjacency graphs (right).

test split is provided with 6,900 and 1,725 designs respectively. We now briefly outline the sketch and extrude modeling operations and provide additional details in Section A.1 of the appendix.

**Sketch**  Unlike free-form sketches, sketches in CAD are composed of 2D geometric primitives (lines, circles, splines etc.), associated dimensions (distance, diameter, angle etc.) and constraints (symmetry, tangent, parallel etc.). Sketch geometry is represented by points, that create curves, that in turn form loops within profiles. The intersection of curves, as the user draws, automatically creates closed loops and profiles that are serialized as both raw curves and trimmed profiles. Sketch profiles form the basis for 3D extrusion as shown in (Figure 2).

**Extrude**  An extrude operation takes one or more sketch profiles and extrudes them from 2D into a 3D B-Rep body. A distance parameter defines how far the profile is extruded. A notable feature of extrude operations in Fusion 360 is the ability to perform Boolean operations in the same step. As a user extrudes a sketch profile, they choose to create a *new body*, *join*, *cut*, or *intersect* with other bodies in the design (Figure 2). Additional extrude options are available such as two-sided extrude, symmetrical extrude, and tapered extrude. The combination of expressive sketches and extrude operations with built in Boolean capability enables a wide variety of designs to be constructed.

## 4 FUSION 360 GYM

Together with the dataset we provide an open source environment, called the *Fusion 360 Gym*, for standardizing the CAD reconstruction task. The *Fusion 360 Gym* wraps the underlying Fusion 360 Python API (Autodesk, 2014) and serves as the environment that interacts with an intelligent agent for the task of CAD reconstruction (Figure 3). Specifically, the *Fusion 360 Gym* formalizes the following Markov Decision Process:

- **state**: Contains the current geometry, and optionally, the target geometry to be reconstructed. We use a B-Rep face-adjacency graph as our state representation.
- **action**: A modeling operation that allows the agent to modify the current geometry. We consider two action representations: sketch extrusion and face extrusion.
- **transition**: *Fusion 360 Gym* implements the transition function that applies the modeling operation to the current geometry.
- **reward**: The user can define custom reward functions depending on the task. We provide intersection over union (IoU) as one measure to compare the current and target geometry.

### 4.1 STATE REPRESENTATION

In order for an agent to successfully reconstruct the target geometry, it is important that we have a suitable state representation. In *Fusion 360 Gym*, we use a similar encoding scheme to Jayaraman et al. (2020) and represent the current and target geometry with a B-Rep face-adjacency graph (Ansaldi et al., 1985), illustrated in Figure 3, right. Crucial to this encoding are the *geometric*

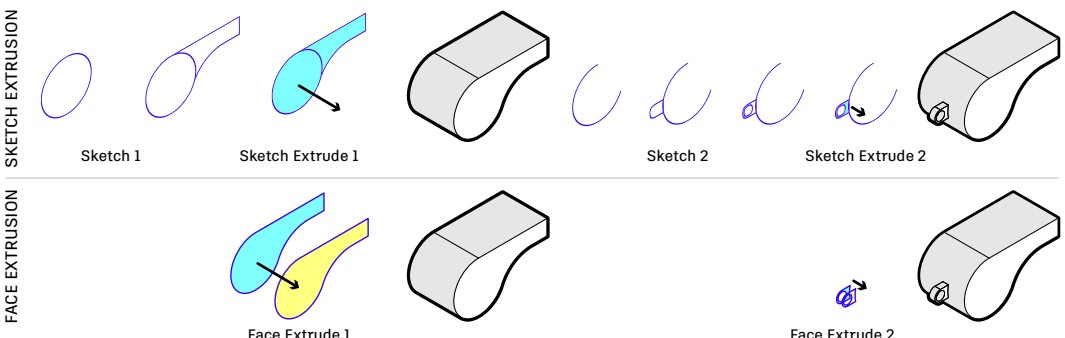

Figure 4: Action representations supported by the *Fusion 360 Gym* include low-level sketch extrusion (top) and simplified face extrusion (bottom).

features of the elements, such as point-locations, and *topological* features specifying how these elements are connected to each other. Specifically, the vertices of the face-adjacency graph represent B-Rep faces (trimmed parametric surfaces) in the design, with graph vertex features representing the size, orientation, and curvature of the faces. The edges of the face-adjacency graph represents B-Rep edges in the design, that connect the adjacent B-Rep faces to each other.

### 4.2 ACTION REPRESENTATION

In the *Fusion 360 Gym* we support two action representations encompassing different modeling operations: *sketch extrusion* and *face extrusion*.

**Sketch Extrusion** In sketch extrusion, the agent must first select a sketch plane, draw on this plane using a sequence of curve primitives, such as lines and arcs, to form closed loop profiles. The agent then selects a profile to extrude a given distance and direction (Figure 4, top). Using this representation it is possible to construct novel geometries by generating the underlying sketch primitives and extruding them by an arbitrary amount. Although all designs in the *Fusion 360 Gallery* reconstruction dataset can be constructed using sketch extrusion, in practice this is challenging. Benko et al. (2002) show that to generate sketches suitable for mechanical engineering parts, the curve primitives often need to be constructed alongside a set of constraints which enforce regularities and symmetries in the design. Although the construction of the constraint graphs is feasible using techniques like the one shown by Liao et al. (2019), enforcing the constraints requires a complex interaction between the machine learning algorithm and a suitable geometric constraint solver, greatly increasing the algorithm complexity. We alleviate this problem by introducing a simplified action representation, called *face extrusion*, that is well suited to learning-based approaches.

**Face Extrusion** In face extrusion, a face from the target design is used as the extrusion profile rather than a sketch profile (Figure 4, bottom). This is possible because the target design is known in advance during reconstruction. An action $a$ in this scheme is a triple $\{\text{face}_{start}, \text{face}_{end}, \text{op}\}$ where the start and end faces are parallel faces referenced from the target geometry, and the operation type is one of the following: *new body, join, cut, intersect*. Target constrained reconstruction using face extrusion has the benefit of narrowly scoping the prediction problem with shorter action sequences and simpler actions. Conversely, not all geometries can be reconstructed with this simplified strategy due to insufficient information in the target, e.g., Extrude 3 in Figure 2 cuts across the entire design without leaving a start or end face. Of the design sequences in the reconstruction dataset, 59.2% can be directly converted to a face extrusion sequence. We estimate that approximately 80% of designs in our dataset can be reconstructed by finding alternative construction sequences.

### 4.3 SYNTHETIC DATA GENERATION

The *Fusion 360 Gym* supports generation of semi-synthetic data by taking existing designs and modifying or recombining them. For instance, we can randomly perturb the sketches and the extrusion distances, and even 'graft' sketches from one design onto another. We also support distribution matching of parameters, such as the number of faces, to ensure that synthetic designs match

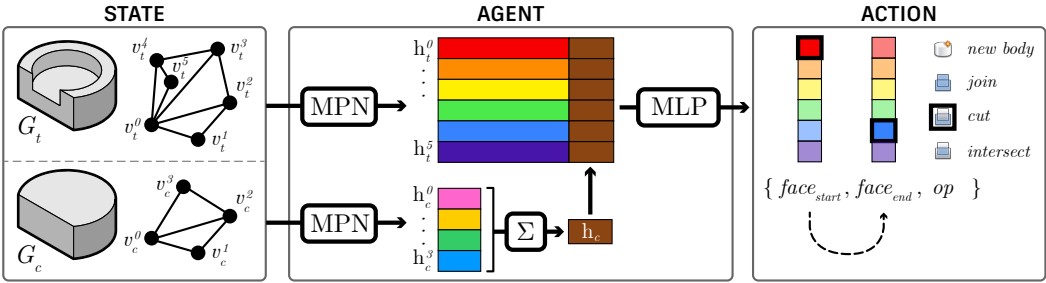

Figure 5: Given a *state* comprising the target geometry $G_t$ and current geometry $G_c$, the *agent* uses a message passing network (MPN) to predict an *action* as a face extrusion modeling operation.

a human-designed dataset distribution. Learning-based systems can leverage semi-synthetic data to expand the number of samples in the *Fusion 360 Gallery* reconstruction dataset. We provide examples of synthetic data and commands for the *Fusion 360 Gym* in Section A.2 of the appendix.

# 5 CAD RECONSTRUCTION

## 5.1 TASK

The goal of CAD reconstruction is to recover the sequence of modeling operations used to construct a CAD model with only the geometry as input. This task can be specified using different input geometry representations, including B-Rep, mesh, or point cloud, with progressively lower fidelity. Each representation presents a realistic scenario where parametric CAD information is absent and needs to be recovered. Given a target geometry $G_t$, we wish to find a sequence of CAD modeling operations (actions) $\mathcal{A} = \{a_0, a_1, \cdots\}$ that generates an output geometry $G$, such that every point in space is in its interior, if and only if, it is also in the interior of $G_t$.

**Evaluation Metrics** We prescribe three evaluation metrics, IoU, exact reconstruction, and conciseness. IoU measures the intersection over union of $G$ and $G_t$: $\texttt{iou}(G, G_t) = |G \cap G_t|/|G \cup G_t|$. Exact reconstruction measures whether $\texttt{iou}(G, G_t) = 1$. As multiple correct sequences of CAD modeling operations exist, a proposed reconstruction sequence $\mathcal{A}$ need not match the ground truth sequence $\hat{\mathcal{A}}_t$ provided an exact reconstruction is found. To measure the quality of exact reconstructions we consider the conciseness of the construction sequence. Let $\texttt{conciseness}(\mathcal{A}, \hat{\mathcal{A}}_t) = |\mathcal{A}|/|\hat{\mathcal{A}}_t|$, where a score $\leq 1$ indicates the agent found an exact reconstruction with equal or fewer steps than the ground truth, and a score $> 1$ indicates more inefficient exact reconstructions.

## 5.2 METHOD

We now present a method for CAD reconstruction using neurally-guided search (Ellis et al., 2019; Kalyan et al., 2018; Tang et al., 2019; Devlin et al., 2017) from *B-Rep input* using *face extrusion* modeling operations. Rather than discovering a sequence of construction by exploration, the agent is trained to match known reconstruction sequences present in the training set using imitation learning. We leverage search at inference time to recover the given target geometry.

**Imitation Learning** To perform imitation learning, we leverage the fact that we have the ground truth sequence of modeling operations (actions) $\hat{\mathcal{A}}_t = \{\hat{a}_{t,0} \cdots \hat{a}_{t,n-1}\}$ for each design $G_t$ in the dataset. We feed the ground truth action sequence $\hat{\mathcal{A}}_t$ into the *Fusion 360 Gym*, starting from the empty geometry $G_0$, and output a sequence of partial constructions $G_{t,1} \cdots G_{t,n}$ where $G_{t,n} = G_t$. We then collect the supervised dataset $\mathcal{D} = \{(G_0, G_t) \rightarrow \hat{a}_{t,0}, (G_{t,1}, G_t) \rightarrow \hat{a}_{t,1} \cdots\}$ and train a supervised agent $\pi_\theta$ that takes the pair of current-target constructions $(G_c, G_t)$ to a modeling operation action $a_c$, which would transform the current geometry closer to the target. Formally, we

optimize the expected log-likelihood of correct actions under the data distribution:

$$E_{(G_c, G_t) \sim \mathcal{D}} \left[ \log \pi_\theta \Big( \hat{a}_c | (G_c, G_t) \Big) \right] \tag{1}$$

**Agent** The agent takes a pair of geometries $(G_c, G_t)$ as state, and outputs the corresponding face-extrusion action $a = \{\text{face}_{start}, \text{face}_{end}, \text{op}\}$ (Figure 5). The two geometries $G_c, G_t$ are given using a face-adjacency graph similar to Jayaraman et al. (2020), where the graph vertexes represent the faces of the geometry, with vertex features calculated from each face: $10 \times 10$ grid of 3D points, normals, and trimming mask, in addition to the face surface type. The edges define connectivity of adjacent faces. Inputs are encoded using two *separate* message passing networks (MPN) aggregating messages along the edges of the graph. The encoded vectors representing the *current* geometry are summed together ($h_c$ in Figure 5), and concatenated with the encoded vertexes of the target geometry $(h_t^0 \cdots h_t^5)$. The concatenated vectors are used to output the action using a multi-layer perceptron (MLP), with the end face conditioned on the vertex embedding of the predicted start face.

**Search** Given a neural agent $\pi_\theta(a | (G_c, G_t))$ capable of furthering a current geometry toward the target geometry, we can amplify its performance at test time using search. Here we report the result of the most basic of search strategies, random rollout, and provide results from additional search strategies in Section A.3 of the appendix. We let the agent interact with the environment by sampling a sequence of actions according to $\pi_\theta$ up to a fixed rollout length of $\max(\frac{f_p}{2}, 2)$, where $f_p$ is the number of planar faces in $G_t$. If the agent is successful at reconstructing the target, we stop. Otherwise, we repeat the process until we exhaust a global search budget.

### 5.3 RESULTS

We now present results on CAD reconstruction using the test set of the *Fusion 360 Gallery* reconstruction dataset. We seek to understand 1) how state-of-the-art baseline networks perform on the CAD reconstruction task, 2) how synthetic data performs compared to human designs. In each experiment we use different agents to reconstruct a target design, while holding the search strategy constant. For a target design $G_t$, each agent uses the random rollout search algorithm and attempts reconstruction over multiple rollouts. Each time the agent takes an action during search (a search step), we track the best IoU the agent has discovered so far, and whether exact reconstruction is achieved. We cap the total search budget to 100 steps.

**Baseline Comparison** We evaluate five different agents to understand how state-of-the-art baseline networks perform on the CAD reconstruction task. The **rand** agent uniformly samples from the available actions to serve as a naive baseline without any learning. **mlp** is a simple MLP agent that does not take advantage of shape topology. **gcn**, **gin**, and **gat** are MPN agents that use a Graph Convolution Network (Kipf & Welling, 2016), Graph Isomorphism Network (Xu et al., 2018), and Graph Attention Network (Veličković et al., 2017) respectively. We use two MPN layers for all comparisons, with standard layer settings as described in the appendix. We report the evaluation

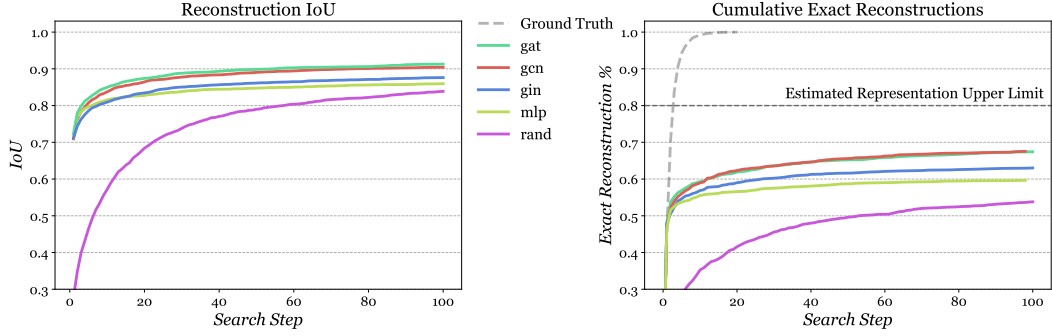

Figure 6: Reconstruction results over 100 search steps using random rollouts with different agents. For exact reconstructions, $0.8$ is the estimated upper limit of the face extrusion action representation.

| Agent | IoU | | Exact Reconstruction % | | Conciseness | # Parameters |
|---|---|---|---|---|---|---|
| | 20 Steps | 100 Steps | 20 Steps | 100 Steps | | |
| gat | **0.8742** | **0.9128** | 0.6191 | 0.6742 | 1.0206 | 3.03M |
| gcn | 0.8644 | 0.9042 | **0.6232** | **0.6754** | 1.0168 | 3.02M |
| gin | 0.8346 | 0.8761 | 0.5901 | 0.6301 | 1.0042 | 3.62M |
| mlp | 0.8274 | 0.8596 | 0.5658 | 0.5965 | **0.9763** | 2.24M |
| rand | 0.6840 | 0.8386 | 0.4157 | 0.5380 | 1.2824 | - |

Table 1: Reconstruction results for IoU and exact reconstruction at 20 and 100 search steps using random rollouts with different agents. Lower values are better for conciseness.

metrics of each agent as a function of the number of steps in Figure 6. We detail the exact results at step 20 and 100 in Table 1. Step 20 represents the point where it is possible to perform exact reconstructions for all designs in the test set. We also detail the conciseness of the recovered sequence for exact reconstructions. We note that all neurally guided agents outperform the random agent baseline. The topology information available with a MPN is found to improve reconstruction performance. The gat and gcn agents show the best performance but fall well short of exact reconstruction on all designs in the test set, demonstrating that the CAD reconstruction task is non-trivial and an open problem for future research.

**Synthetic Data Performance**   We evaluate four gcn agents trained on different data sources to understand how synthetic data performs compared to human design data. **real** is trained on the human design training set. **syn** is trained on synthetic data from procedurally generated sketches of rectangles and circles extruded randomly (Figure 8, left). **semi-syn** is trained on semi-synthetic designs that use existing sketches in the training set with two or more extrude operations to match the distribution of the number of faces in the training set (Figure 8, right). **aug** is trained on the human design training set mixed with additional semi-synthetic data. We hold the training data quantity constant across agents, with the exception of the aug agent that contains a larger quantity from two sources. All agents are evaluated on the human design test set.

Figure 7 shows that training on human design data offers a significant advantage over synthetic and semi-synthetic data. For the aug agent reconstruction performance is aided early on by data augmentation. This is likely due to semi-synthetic designs with 1 or 2 extrusions appearing similar to human designs. Conversely, semi-synthetic designs with multiple randomly applied extrusions appear less and less similar to human design. This difference in distribution between human and synthetic designs becomes more prevalent as search progresses and adversely affects performance.

**Qualitative Results**   Figure 9 shows ground truth construction sequences compared with other agents using random search. The rollout with the highest IoU is shown with the IoU score and total search steps taken. Steps that don't change the geometry or occur after the highest IoU are omitted from the visualization.

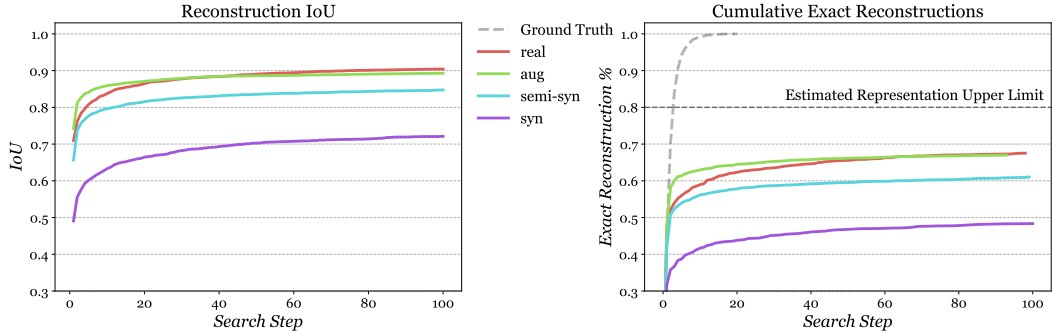

Figure 7: Reconstruction results over 100 search steps using random rollouts and gcn agents trained on human-designed data (real), a mixture of human-designed and semi-synthetic data (aug), semi-synthetic data (semi-syn), and synthetic data (syn).

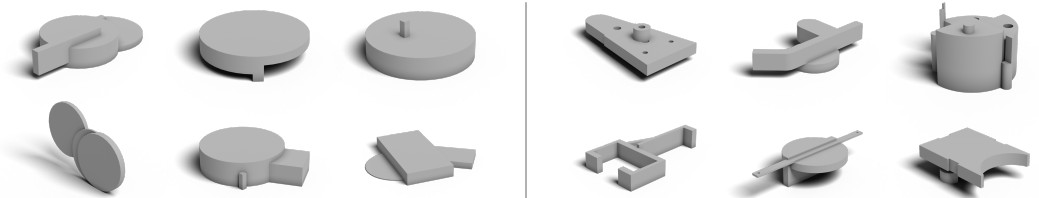

Figure 8: Example synthetic (left) and semi-synthetic data (right).

**Discussion** For practical application of CAD reconstruction it is necessary to have an exact reconstruction where all details of a design are reconstructed in a concise way. It is notable that incorrect reconstructions can score well with the IoU metric, but omit important design details. We therefore suggest IoU should be a secondary metric, with future work focusing on improving exact reconstruction performance with concise construction sequences. Conciseness should always be considered alongside exact reconstruction performance as naive approaches that only reconstruct short sequences can achieve good conciseness scores.

## 6 CONCLUSION AND FUTURE DIRECTIONS

In this paper we presented the *Fusion 360 Gallery* reconstruction dataset and environment for learning CAD reconstruction from sequential 3D CAD data. We outlined a standard CAD reconstruction task, together with evaluation metrics, and presented results from a neurally guided search approach. We envision a number of future directions that could leverage the reconstruction dataset: new representations for sequential geometry capable of performing CAD reconstruction and generation from B-Rep, mesh, point cloud, or image data; reinforcement learning approaches that mimic and improvise sequential modeling operations; and sketch and constraint synthesis from 3D geometry or images. Finally, beyond the simplified design space of sketch and extrude lies the full breadth of rich sequential CAD modeling operations.

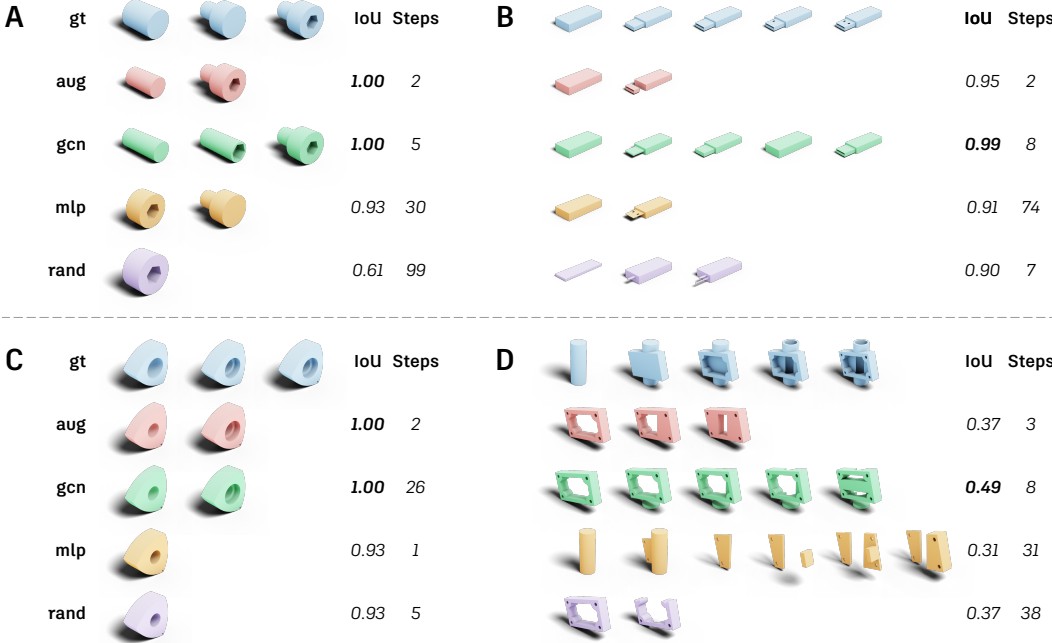

Figure 9: Qualitative construction sequence results comparing the ground truth (gt) to reconstructions using different agents with random search.

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

## A   APPENDIX

### A.1   FUSION 360 GALLERY RECONSTRUCTION DATASET

In this section we provide additional details on the *Fusion 360 Gallery* reconstruction dataset.

#### A.1.1   DATA PROCESSING

We acquire the Fusion 360 designs from the Autodesk Online Gallery[2]. From the approximately 20,000 designs available we derive several datasets focused on specific areas of research. This paper introduces the *reconstruction dataset* as a new baseline dataset for CAD reconstruction. We use the Fusion 360 Python API to parse the native .f3d files. We divide multi-component assemblies into separate designs and suppress modeling operations other than *sketch* and *extrude* to expand the data quantity. Figure 10 shows an example assembly that is split up to produce multiple designs with independent construction sequences. The rounded edges are removed by suppressing fillets in the parametric CAD file.

After each construction sequence has been extracted we perform reconstruction and compare the reconstructed design to the original to ensure data validity. Failure cases and any duplicate designs, are not included in the dataset. We consider a design a duplicate when there is an exact match in all of the following: body count, face count, surface area to one decimal point, volume to one decimal point, and for each extrude in the construction sequence: extrude profile count, extrude body count, extrude face count, extrude side face count, extrude end face count, and extrude start face count. This process allows us to match designs that have been translated or rotated, while considering designs unique if they have matching geometry but different construction sequences. Deduplication removes approximately 5,000 designs. Figure 11 shows a random sampling of designs from the reconstruction dataset.

#### A.1.2   GEOMETRY DATA FORMAT

We provide geometry in several data formats that we discuss in this section. Geometry is provided for the final design and after each extrude operation.

**Boundary Representation**   A B-Rep consists of faces, edges, loops, coedges and vertices (Weiler, 1986). A face is a connected region of the model's surface. An edge defines the curve where two faces meet and a vertex defines the point where edges meet. Faces have an underlying parametric surface which is divided into visible and hidden regions by a series of boundary loops. A set of connected faces forms a body. Designs in the dataset may contain multiple bodies.

---

[2]https://gallery.autodesk.com

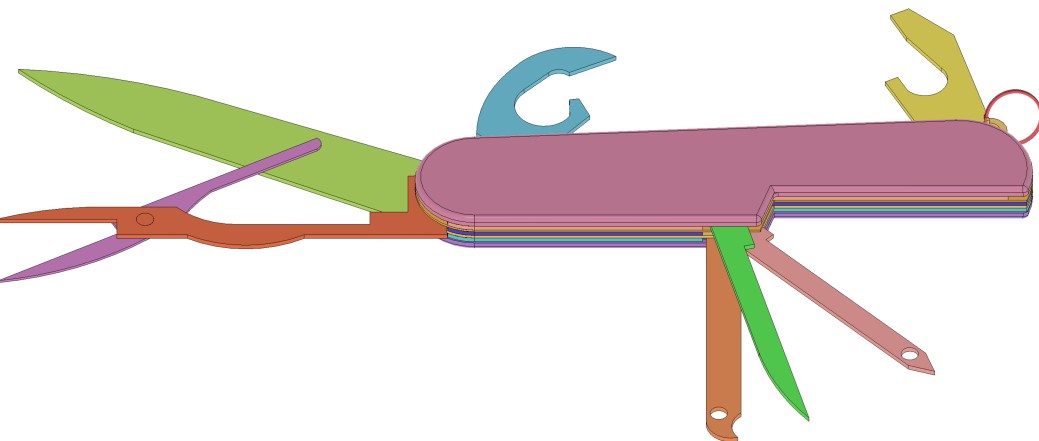

Figure 10: An example multi-component assembly that is broken up into separate designs (highlighted with color), each with an independent construction sequence.

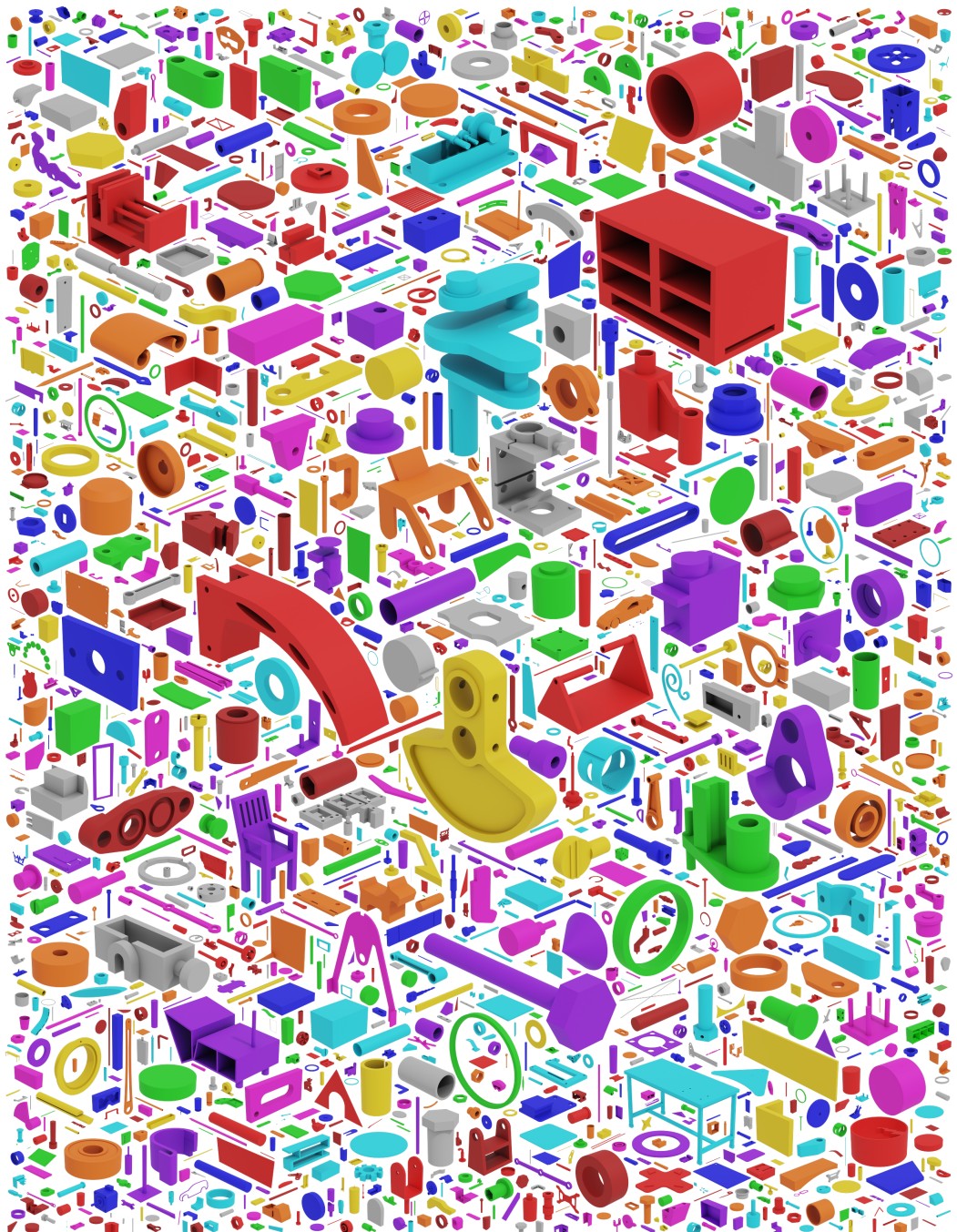

Figure 11: A random sampling of designs from the *Fusion 360 Gallery* reconstruction dataset.

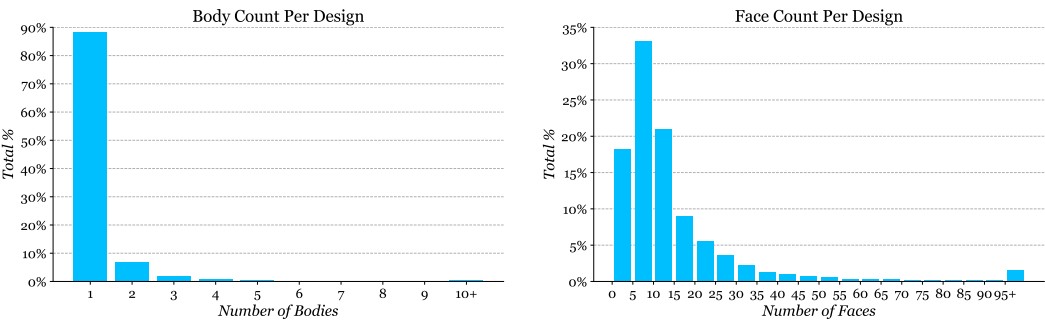

Figure 12: Left: The number of bodies per design shown as a distribution. Right: The number of B-Rep faces per design shown as a distribution.

B-Rep data is provided as .smt files representing the ground truth geometry and .step as an alternate neutral B-Rep file format. The .smt file format is the native format used by Autodesk Shape Manager, the CAD kernel within Fusion 360, and has the advantage of minimizing conversion errors. Additionally the B-Rep entities, such as bodies and faces, can referenced from the construction sequence back to entities in the .smt file.

**Mesh**  Mesh data is provided in .obj format representing a triangulated version of the B-Rep. Each B-Rep face is triangulated separately and labeled as a group of triangles in the .obj file with the B-Rep face id as the group name. This approach allows the triangles to be traced back to the B-Rep face and associated extrude operation. Note that the .obj meshes provided are not manifold.

Other representations, such as point clouds or voxels, can be generated using existing data conversion routines and are not included in the dataset. For convenience we include a thumbnail .png image file together with each geometry.

Files are provided in a single directory, with a naming convention as follows: `XXXXX_YYYYYYY_ZZZZ[_1234].ext`. Here `XXXXX` represents the project, `YYYYYYY` the file, `ZZZZ` the component, and `_1234` the extrude index. If `_1234` is absent the file represents the final design.

### A.1.3   Design Complexity

A key goal of the reconstruction dataset is to provide a suitably scoped baseline for learning-based approaches to CAD reconstruction. Restricting the modeling operations to *sketch* and *extrude* vastly narrows the design space and enables simpler shape grammars for reconstruction. Each design represents a component in Fusion 360 that can have multiple geometric bodies. Figure 12 (left) illustrates that the vast majority of designs have a single body. The number of B-Rep faces in each design gives a good indication of the complexity of the dataset. Figure 12 (right) shows the number of faces per design as a distribution, with the peak being between 5-10 faces per design. As we do not filter any of the designs based on complexity, this distribution reflects real designs where simple washers and flat plates are common components in mechanical assemblies.

### A.1.4   Construction Sequence

The construction sequence is the series of *sketch* and *extrude* operations that are executed to produce the final geometry. We provide the construction sequence in a JSON format text file. Each step in the construction sequence has associated parameters that are stored in that entity. For example, *sketch* entities will store the curves that make up the sketch. Each construction sequence must have at least one *sketch* and one *extrude* step, for a minimum of two steps. The average number of steps is 4.74, the median 4, the mode 2, and the maximum 61. Figure 13 illustrates the distribution of construction sequence length and the most frequent construction sequence combinations.

With access to the full parametric history, it is possible to extract numerous relationships from the dataset that can be used for learning. Starting at a high level, we know the order of modeling operations in the construction sequence. The sketch geometry, B-Rep faces, and triangles derived from

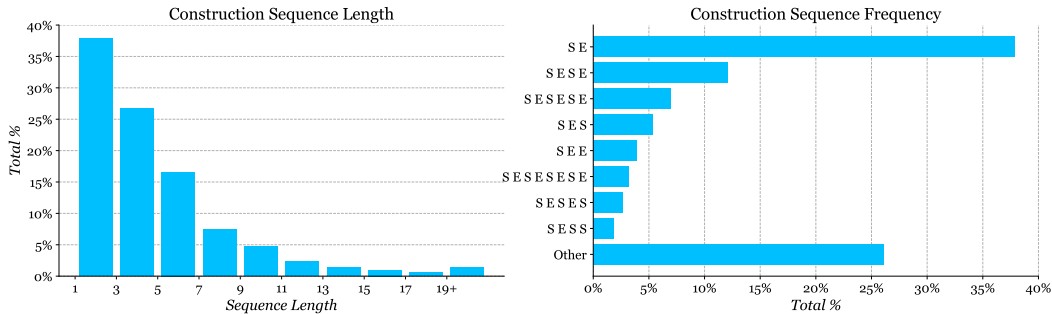

Figure 13: Left: The distribution of construction sequence length. Right: The distribution of common construction sequences. S indicates a *Sketch* and E indicates an *Extrude* operation.

them, can be traced back to a position in the construction sequence. The type of geometry created by each modeling operation is also known. For example, sketches create trimmed profiles where the curves intersect to form closed loops; extrude operations produce B-Rep faces with information such as which faces were on the side or ends of an extrusion. In addition, the sequence of B-Rep models themselves contain valuable topology information that can be leveraged, such as the connectivity of B-Rep faces and edges. Finally geometric information like points and normal vectors can be sampled from the parametric surfaces. Feature diversity enables many different learning representations and architectures to be leveraged and compared.

### A.1.5 SKETCH

In this section we describe the sketch data in further detail and present statistics illustrating the data distribution. Figure 14 illustrates the geometric 2D primitives, described in section 3, that make up a sketch. Sketches are represented as a series of points ($pt1...pt6$), that create curves ($c1...c5$), that in turn create profiles ($pr1...pr3$), illustrated with separate colors. Profiles can have inner loops to create holes, $c1$ is the inner loop of $pr2$ and the outer loop of $pr3$. Profiles also have a trimmed representation that contains only closed loops without open curves. The trimmed representation is shown in the lower right of Figure 14 where the $c5$ is trimmed and incorporated into $pr1$ and $pr2$.

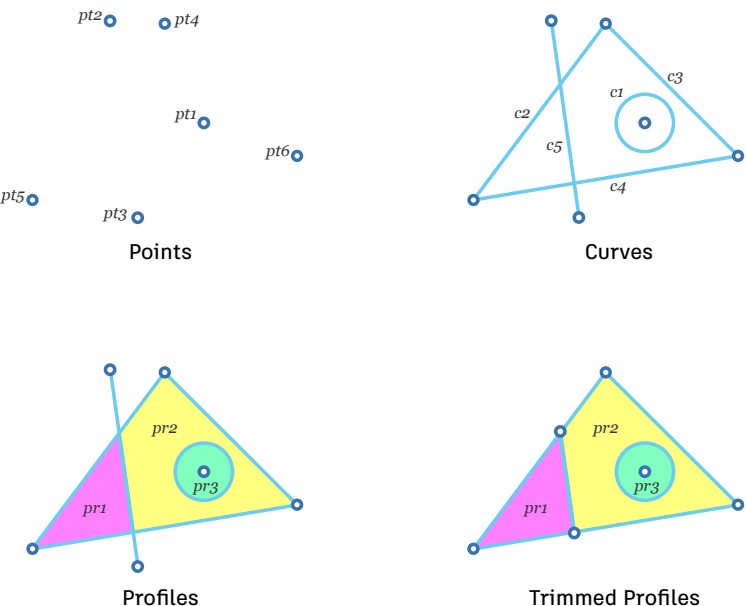

Figure 14: Sketch primitives.

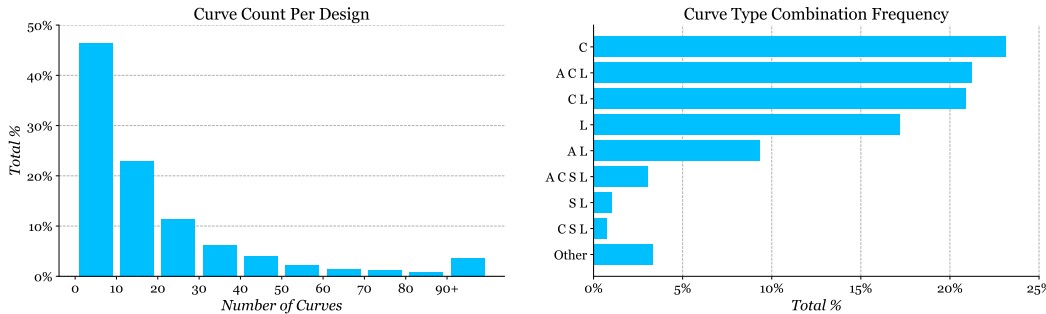

Figure 15: Left: The number of curves in each design, shown as a distribution. Right: Common curve combinations in each design, shown as a distribution. Each curve type is abbreviated as follows: C - *SketchCircle*, A - *SketchArc*, L - *SketchLine*, S - *SketchFittedSpline*.

All sketch geometry is provided in a solved state, meaning a sketch constraint solver is not required for standard reconstruction. The as-designed ordering of sketch operations is not stored in the native design files, however a consistent ordering can be derived by traversing the sketch profiles in sequence.

**Points**   Each point is provided with a universally unique identifier (UUID) key and a `Point3D` data structure with $x$, $y$, and $z$. Sketch primitives are drawn in a local 2D coordinate system and later transformed into world coordinates. As such all sketch points have a $z$ value of 0.

**Curves**   Each curve has a UUID key and a `SketchCurve` that can represent the curve types listed below. The parameters for each curve type can be referenced via the Fusion 360 API documentation linked below.

- `SketchArc`
- `SketchCircle`
- `SketchConicCurve`
- `SketchEllipse`
- `SketchEllipticalArc`
- `SketchFittedSpline`
- `SketchFixedSpline`
- `SketchLine`

Figure 15 illustrates the distribution of curve count per design and the frequency that different curve combinations are used together in a design. It is notable that mechanical CAD sketches rely heavily on lines, circles, and arcs rather than spline curves.

**Profiles**   Profiles represent a collection of curves that join together to make a closed loop. In *Fusion 360* profiles are automatically generated from arbitrary curves that don't necessarily connect at the end points. In Figure 14 two profiles ($pr1$ and $pr2$) are generated when the line crosses the triangle. We provide both the original curves (Figure 14, top right) used to generate the profiles (Figure 14, bottom left) and the trimmed profile information containing just the closed profile loop (Figure 14, bottom right). Loops within profiles have a flag that can be set to specify holes.

**Dimensions**   User specified sketch dimensions are used to define set angles, diameters, distances etc. between sketch geometry to constraint the sketch as it is edited. Each dimension has a UUID key and a `SketchDimension` that can represent the dimension types listed below. Each dimension references one or more curves by UUID. The parameters for each dimension type can be referenced via the Fusion 360 API documentation linked below.

- `SketchAngularDimension`
- `SketchConcentricCircleDimension`

- `SketchDiameterDimension`
- `SketchEllipseMajorRadiusDimension`
- `SketchEllipseMinorRadiusDimension`
- `SketchLinearDimension`
- `SketchOffsetCurvesDimension`
- `SketchOffsetDimension`
- `SketchRadialDimension`

**Constraints**   Constraints define geometric relationships between sketch geometry. For example, a symmetry constraint enables the user to have geometry mirrored, or a parallel constraint ensures two lines are always parallel. Each constraint has a UUID key and a `GeometricConstraint` that can represent the constraint types listed below. Each constraint references one or more curves by UUID. The parameters for each constraint type can be referenced via the Fusion 360 API documentation linked below.

- `CircularPatternConstraint`
- `CoincidentConstraint`
- `CollinearConstraint`
- `ConcentricConstraint`
- `EqualConstraint`
- `HorizontalConstraint`
- `HorizontalPointsConstraint`
- `MidPointConstraint`
- `OffsetConstraint`
- `ParallelConstraint`
- `PerpendicularConstraint`
- `PolygonConstraint`
- `RectangularPatternConstraint`
- `SmoothConstraint`
- `SymmetryConstraint`
- `TangentConstraint`
- `VerticalConstraint`
- `VerticalPointsConstraint`

Figure 16 illustrates the distribution of dimension and constraint types in the dataset.

### A.1.6   EXTRUDE

In this section we describe the extrude data in further detail and present statistics illustrating the data distribution. Extrude operations have a number of parameters that are set by the user while designing. Figure 17 shows how a sketch (left) can be extruded a set distance on one side, symmetrically

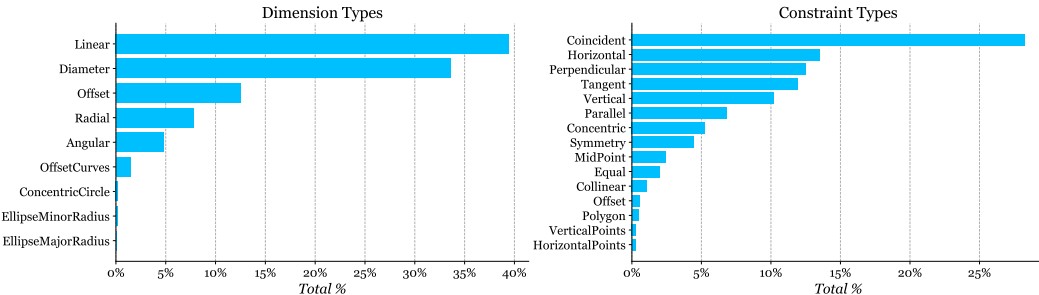

Figure 16: The distribution of constraint (left) and dimension (right) types.

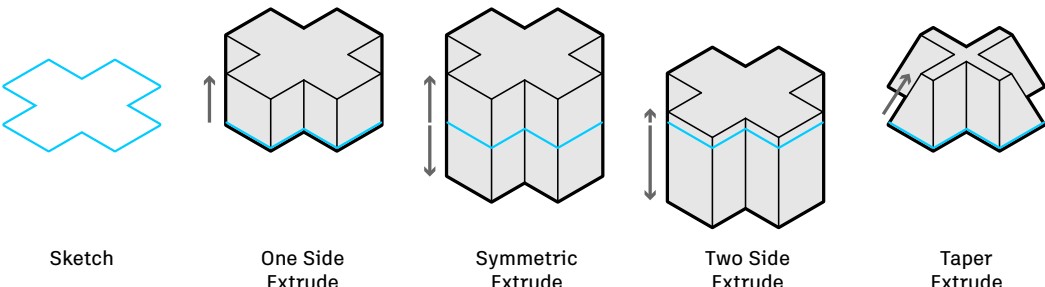

Figure 17: An extrude can be expressed in several different ways: perpendicular from a sketch for a set distance along one side, a symmetrical distance along both sides, or separate distances along two sides. Additionally the extrude can be tapered at an angle.

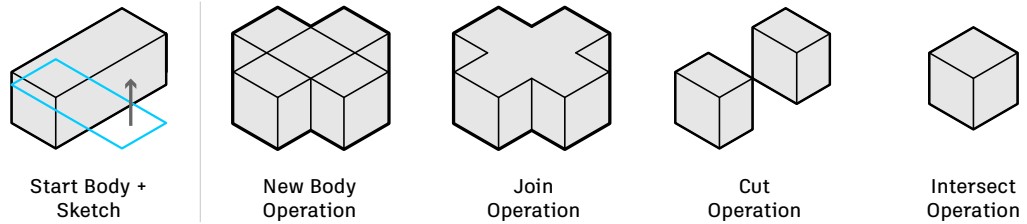

Figure 18: Extrude operations include the ability to Boolean with other geometry. From the starting body shown on the left, a sketch is extruded to form a new body overlapping the starting body, joined with the starting body, cut out of the starting body, or intersected with the starting body.

on two sides, with different distances on each side, as well as tapered. Once a single body has been created by an extrude operation, subsequent extrudes can interact with that body via Boolean operations. Figure 18 shows a starting body and sketch (left) that can be extruded to form two separate bodies, a single joined body, a cut through the starting body, or a body at the intersection. The first extrude operation of a construction sequence is always a new body, with any operation possible for subsequent operations.

Figure 19 outlines the distribution of different extrude types and operations. Note that tapers can be applied in addition to any extrude type, so the overall frequency of each is shown rather than a relative percentage.

## A.2 FUSION 360 GYM

In this section we provide additional information about the functionality available in the *Fusion 360 Gym*. The *Fusion 360 Gym* requires the Autodesk Fusion 360 desktop CAD application, available on both macOS and Windows for free to the academic community. Although Fusion 360 is a cloud

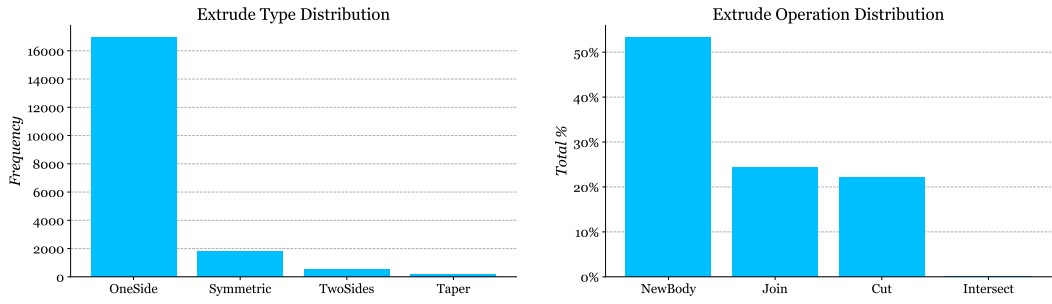

Figure 19: The distribution of extrude types (left) and operations (right).

connected desktop application, the *Fusion 360 Gym* does all processing locally. The *Fusion 360 Gym* consists of a *server* that runs inside of Fusion 360 and receives commands from a *client* running externally. Multiple instances of the *Fusion 360 Gym server* can be run in parallel. The remainder of this section introduces the available commands from the *client*.

### A.2.1 RECONSTRUCTION COMMANDS

Reconstruction commands can reconstruct the existing designs at different granularity levels from json files provided with the *Fusion 360 Gallery* reconstruction dataset.

- `reconstruct(file)`: reconstruct an entire design from the provided json file.
- `reconstruct_sketch(json_data, sketch_name, sketch_plane, scale, translate, rotate)`: reconstruct a sketch from the provided json data and a sketch name. A `sketch_plane` can be either: (1) a string value representing a construction plane: `XY`, `XZ`, or `YZ`; (2) a B-Rep planar face id; or (3) a point3d on a planar face of a B-Rep.
- `reconstruct_profile(json_data, sketch_name, profile_id, scale, translate, rotate)`: reconstruct a profile from the provide json data, a sketch name, and a profile id.
- `reconstruct_curve(json_data, sketch_name, curve_id, scale, translate, rotate)`: reconstruct a curve from the provide json data, a sketch name, and a curve id.
- `set_target(file)`: set the target to be reconstructed with a .step or .smt file. The call returns a face adjacency graph representing the B-Rep geometry/topology and a *bounding_box* of the target that can be used for normalization.
- `revert_to_target()`: revert to the target design, removing all reconstruction geometry.

### A.2.2 SKETCH EXTRUSION COMMANDS

Sketch extrusion commands allows users to incrementally create new designs by generating the underlying sketch primitives and extruding them by an arbitrary amount.

- `add_sketch(sketch_plane)`: add a sketch to the design. A `sketch_plane` can be either: (1) a string value representing a construction plane: `XY`, `XZ`, or `YZ`; (2) a B-Rep planar face id; or (3) a point3d on a planar face of a B-Rep.
- `add_point(sketch_name, p1, transform)`: add a point to create a new sequential line in the given sketch. `p1` is either a point in the 2D sketch space or a point in the 3D world coordinate space if `transform="world"` is specified.
- `add_line(sketch_name, p1, p2, transform)`: add a line to the given sketch. `p1` and `p2` are the same as defined in `add_point()`.
- `add_curve(sketch_name, curve_data, transform)`: add a curve to the given sketch. `curve_data` follows the format supplied to `reconstruct_curve()`.
- `close_profile(sketch_name)`: close the current set of lines to create one or more profiles by joining the first point to the last point.
- `add_extrude(sketch_name, profile_id, distance, operation, export_type, is_IoU)`: add an extrude to the design. Four operations are supported: `JoinFeatureOperation`, `CutFeatureOperation`, `IntersectFeatureOperation`, or `NewBodyFeatureOperation`. Two export formats are provided: (1) `BRep` to represent B-Rep vertices of the resulting body and B-Rep face information; (2) `Graph` to represent a face adjacency graph representing the B-Rep geometry/topology. An intersection over union (IoU) value between the target and the reconstruction is calculated and returned if `is_IoU="True"` is specified.

### A.2.3 FACE EXTRUSION COMMANDS

Face extrusion commands enable a target design to be reconstructed using extrude operations from face to face.

- `add_extrude_by_target_face(start_face, end_face, operation)`: add an extrude between two faces of the target. Four operations are supported: `JoinFeatureOperation`, `CutFeatureOperation`, `IntersectFeatureOperation`, or `NewBodyFeatureOperation`.

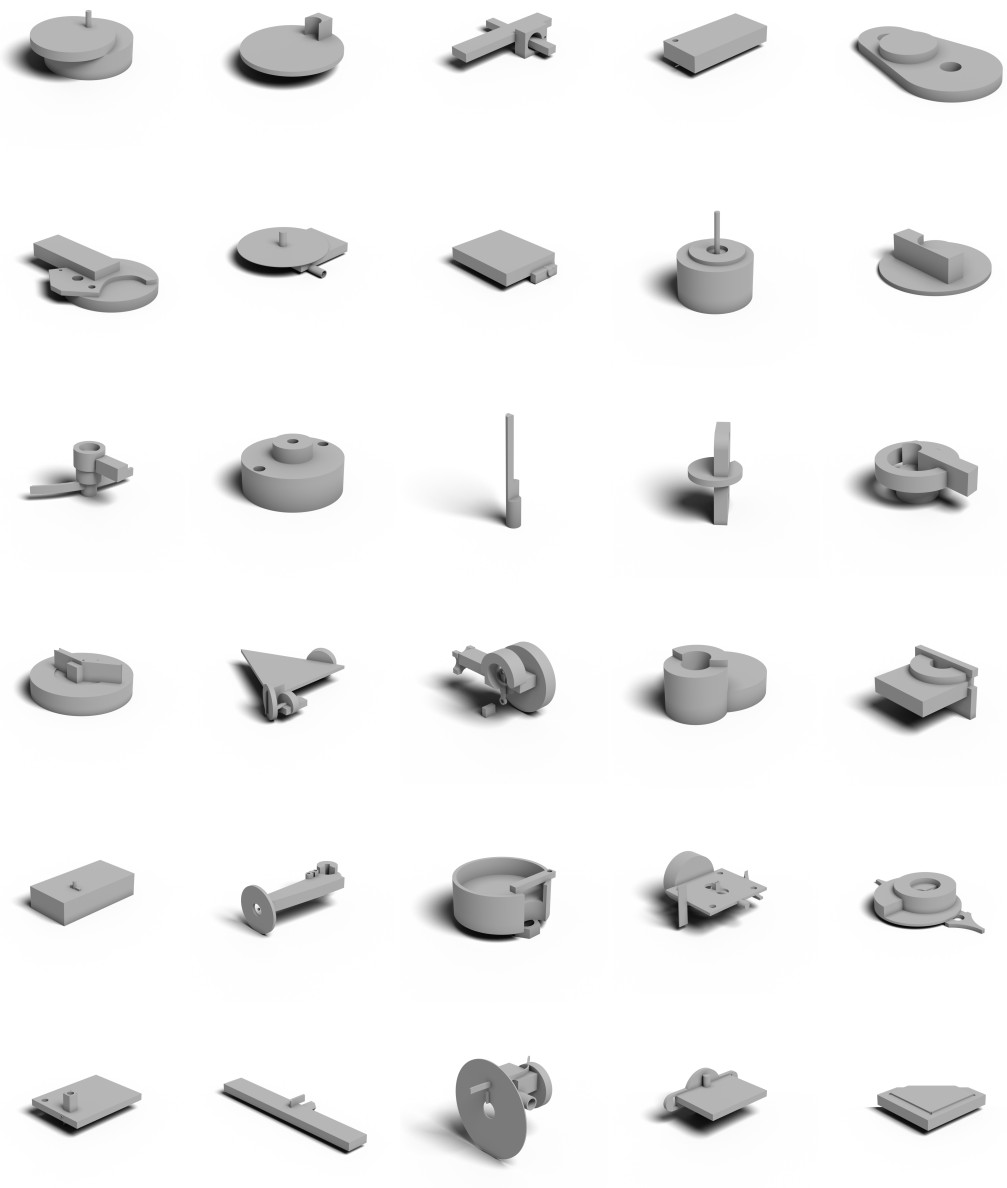

Figure 20: Example designs created using randomized reconstruction commands.

- `add_extrudes_by_target_face(actions, revert)`: execute multiple extrude operations, between two faces of the target, in sequence.

### A.2.4 RANDOMIZED RECONSTRUCTION COMMANDS

Randomized reconstruction commands allow users to sample designs, sketches, and profiles from existing designs in the *Fusion 360 Gallery* and support distribution matching of parameters, in support of generations of semi-synthetic data. Figure 20 shows example designs created using randomized reconstruction commands.

- `get_distributions(data_dir, filter)`: get a list of distributions from the provided dataset. The command currently supports the following distributions: `the starting sketch place`, `the`

```
number of faces, the number of extrusions, the length of sequences, the
number of curves, the number of bodies, the sketch areas, and the profile
areas.
```

- `distribution_sampling(distributions, parameters)`: sample distribution matching parameters for one design from the distributions.

- `sample_design(data_dir)`: randomly sample a json file from the given dataset.

- `sample_sketch(json_file, sampling_type, area_distribution)`: sample one sketch from the provided design. Three sampling types are provided: (1) `random`, return a sketch randomly sampled from the provided design; (2) `deterministic`, return the largest sketch in the design; and (3) `distributive`, return a sketch that its area is in the distribution of the provided dataset.

- `sample_profiles(sketch_name, max_number_profiles, sampling_type, area_distribution)`: sample profiles from the provided sketch. Three sampling types are provided: (1) `random`, return profiles randomly sampled from the provided sketch; (2) `deterministic`, return profiles that are larger than the average area of the profiles in the sketch; and (3) `distributive`, return profiles that the areas are in the distribution of the provided dataset.

### A.2.5 EXPORT COMMANDS

Export commands enable the existing designs to be exported in the following formats:

- `mesh(file)`: retrieve a mesh in .obj or .stl format and write it to the local file provided.

- `brep(file)`: retrieve a brep in .step, .smt, or .f3d format and write it to a local file provided.

- `sketches(dir, format)`: retrieve each sketch in .png or .dxf format and write them to a local directory provided.

- `screenshot(file, width, height)`: retrieve a screenshot of the current design as a png image and write it to a local file provided.

- `graph(file, dir, format)`: retrieve a face adjacency graph in a given format and write it in a local directory provided.

### A.3 CAD RECONSTRUCTION

In this section we provide additional details of the experiments performed on the CAD reconstruction task described in Section 5.

### A.3.1 DATA PREPARATION

The agents are trained on a subset of the reconstruction dataset that has been converted into a face extrusion sequence. Due to the simplified face extrusion representation, not all designs from the dataset can be converted to a face extrusion sequence. Figure 21 shows several common conversion limitations where necessary face information (highlighted in red) is not present in the target geometry. The intermediate top face in Figure 21 B disappears when merged with the top face of Extrude 2. In Figure 21 C a hole cut through the geometry means the intermediate top face of Extrude 1 is absent and there is no start or end face in the target geometry to perform the cut operation used in Extrude 2. Although it is possible to find alternate face extrusion sequences with heuristic rules, we instead try to maintain the user designed sequence with the exception of reversing the direction of the extrusion in some scenarios, e.g. the end face becomes the start face.

### A.3.2 AGENT

All MPN agents employ a network architecture able to exploit the graph structure of the data, consisting of two layers passing messages along the edges of the graph. The vertex features in the face-adjacency graph are as follows:

- *Points*: A 10×10 grid of 3D points sampled from the UV coordinate space of the B-Rep face and normalized to the bounding box of the target geometry.

- *Normals*: A 10×10 grid of 3D normal vectors sampled from the UV coordinate space of the B-Rep face.

- *Trimming Mask*: A 10×10 grid of binary values representing samples that are inside/outside the B-Rep face trimming boundary.

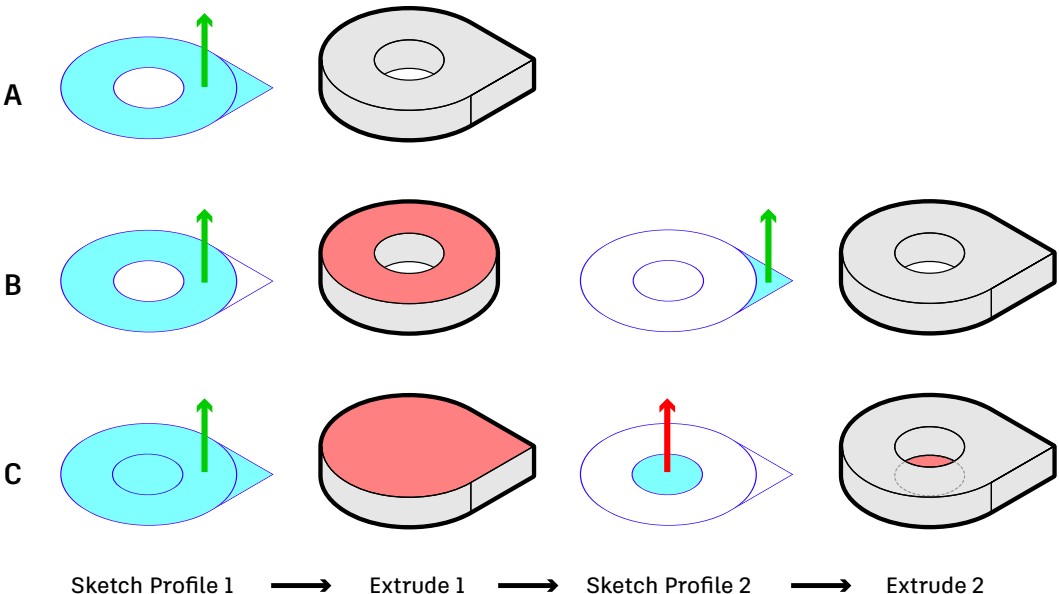

Figure 21: Different construction sequences (A-C) for the same geometry. During conversion to a face extrusion sequence, the necessary face information (highlighted in red) does not exist in the target, meaning B and C can not be converted. Green arrows indicate *new body/join* extrude operations, while red arrows indicate *cut* extrude operations.

- *Surface Type*: A one-hot encoded flag indicating the type of surface represented by the B-Rep face: `Cone`, `Cylinder`, `Elliptical`, `EllipticalCylinder`, `Nurbs`, `Plane`, `Sphere`, `Torus`.

We denote the learned vertex embedding vectors produced by the two MPN branches as $\{\mathbf{h}_c^i\}$ and $\{\mathbf{h}_t^j\}$ for the current output and target graphs, respectively. We estimate the probability of the $k$-th operation type, and the $j$-th face being the start face or end face as:

$$P_{op}^k = F_{op}(\mathbf{h}_c), \ \ \mathbf{h}_c = \sum_i \mathbf{h}_c^i \tag{2}$$

$$P_{start}^j = \mathtt{softmax}_j\Big(F_{start}\big(\mathbf{h}_t^j, \mathbf{h}_c\big)\Big) \tag{3}$$

$$P_{end}^j = \mathtt{softmax}_j\Big(F_{end}\big(\mathbf{h}_t^j, \mathbf{h}_t^{\tilde{j}}, \mathbf{h}_c\big)\Big), \ \ s.t. \ \tilde{j} = \mathtt{argmax}_j P_{start}^j \tag{4}$$

where $F_{op}$, $F_{start}$, and $F_{end}$ denote linear layers that take the concatenated vectors as input.

Using the face extrusion sequence data, we train the agents in an offline manner without interacting with the *Fusion 360 Gym*. The **mlp** and **gcn** agents have a hidden dimension of 256 across all layers. The **gin** agent has two 256-dimensional linear layers within its graph convolution layer. The **gat** has 8 heads of 64 hidden dimensions each. The agents are trained with a dropout rate of 0.1 and a learning rate of 0.0001 for 100 epochs with the model saved at the lowest training loss. The learning rate is decreased by a factor of 0.1 upon plateau within 10 most recent epochs. Training is performed on an NVIDIA Tesla V100 with an Adam optimizer and takes approximately 6-8 hours.

### A.3.3 SEARCH

In addition to the random rollout search (**rand**) described in Section 5.2 we implement beam search (**beam**) and best first search (**best**). Beam search explores multiple candidates in parallel, filtering the top-k candidates, ranked by the generation probability, until a certain length. Best search explores the search space by expanding the most-likely sequence, also ranked by the generation probability.

In all search algorithms we mask out the following invalid actions so they are never taken:

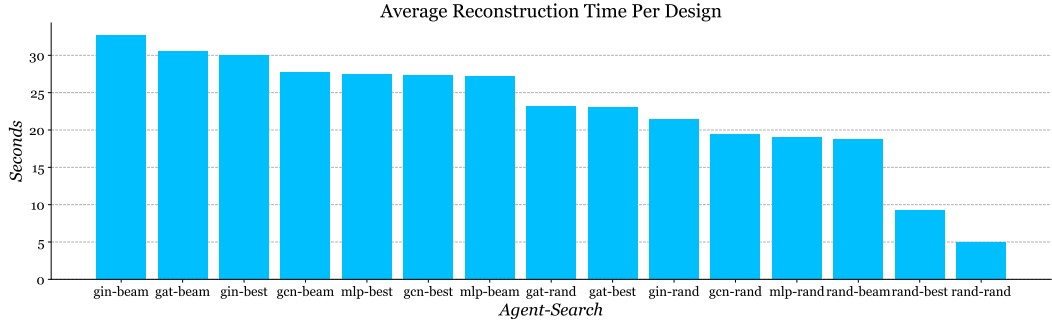

Figure 22: Average reconstruction time per design for combinations of agents and search strategies.

- Start faces surface types that are non-planar
- End faces surface types that are non-planar
- Operation types other than *new body* when the current geometry is empty

Other invalid actions that require geometric checks, such as specifying a start face and end face that are co-planar, are returned as invalid from the *Fusion 360 Gym* and count against the search budget.

### A.3.4 EVALUATION

We perform evaluation using the official test set containing 1725 designs. Evaluation is performed in an online manner using the *Fusion 360 Gym*. Figure 22 shows the average reconstruction time for each design with combinations of agents and search strategies. Time differences are due in part to the number of invalid actions chosen by an agent that can be quickly checked in the *Fusion 360 Gym* without geometry processing. The large majority of evaluation time is spent inside the *Fusion 360 Gym* executing modeling operations, graph generation, and IoU calculation. We set a hard time limit of 10 minutes per design, after which we halt search, affecting between 0-14 designs depending on the agent and search strategy. Between 0-15 designs cause software crashes. 17 designs in the test set cannot be represented as graphs due to our data pipeline not supporting edges with more than two adjacent faces. In all failure cases we use the best seen IoU, or 0 if no IoU score is available, and consider the design to fail at exact reconstruction.

### A.3.5 RESULTS

Table 2 details the full set of results for all agents and search strategies in the extendedbaseline comparison experiment from Section 5.3. Table 3 provides additional details of the synthetic data performance experiment from Section 5.3. Figure 23 visually compares three different search strategies side by side using the gcn agent. We observe that all search strategies perform similarly for reconstruction IoU, while random rollout search has a notable advantage in exact reconstructions. This advantage is due to the limited search budget we enforce to reflect a real-world scenario. We expect both best and beam search to improve with larger search budgets.

### A.4 TASKS

In addition to CAD reconstruction the *Fusion 360 Gallery* reconstruction dataset and *Fusion 360 Gym* can be used for a range tasks such as program synthesis, sequence modeling, generative models, and geometric deep learning. Other tasks include:

- *Modeling operation order prediction* to recover the correct order of construction from raw geometry.
- *Sketch synthesis* to recover the original sketch, including constraints and dimensions, from the 3D geometry.
- *Predicting next action* in the design sequence for 'CAD autocomplete'.
- *Generative models* that are aware of the design sequence and constraints.

| Agent | Search | IoU | | Exact Reconstruction. % | | Conciseness | # Parameters. |
|-------|--------|----------|-----------|----------|-----------|-------------|---------------|
| | | 20 Steps | 100 Steps | 20 Steps | 100 Steps | | |
| gcn | rand | 0.8644 | 0.9042 | **0.6232** | **0.6754** | 1.0168 | 3.02M |
| gcn | beam | 0.8640 | 0.8982 | 0.5739 | 0.6122 | 0.9275 | 3.02M |
| gcn | best | 0.8831 | **0.9186** | 0.5971 | 0.6348 | 0.9215 | 3.02M |
| mlp | rand | 0.8274 | 0.8596 | 0.5658 | 0.5965 | 0.9763 | 2.24M |
| mlp | beam | 0.8619 | 0.8995 | 0.5525 | 0.5884 | 0.9271 | 2.24M |
| mlp | best | 0.8712 | 0.8991 | 0.5675 | 0.5977 | 0.9305 | 2.24M |
| gat | rand | 0.8742 | 0.9128 | 0.6191 | 0.6742 | 1.0206 | 3.03M |
| gat | beam | 0.8691 | 0.9016 | 0.5791 | 0.6133 | 0.9261 | 3.03M |
| gat | best | **0.8895** | 0.9139 | 0.5994 | 0.6354 | 0.9290 | 3.03M |
| gin | rand | 0.8346 | 0.8761 | 0.5901 | 0.6301 | 1.0042 | 3.62M |
| gin | beam | 0.8500 | 0.8913 | 0.5594 | 0.5983 | 0.9299 | 3.62M |
| gin | best | 0.8693 | 0.9007 | 0.5803 | 0.6122 | 0.9340 | 3.62M |
| rand | rand | 0.6840 | 0.8386 | 0.4157 | 0.5380 | 1.2824 | - |
| rand | beam | 0.4785 | 0.6277 | 0.2812 | 0.3896 | 0.9118 | - |
| rand | best | 0.6334 | 0.7994 | 0.3693 | 0.4887 | **0.8979** | - |

Table 2: Reconstruction results for multiple agent and search combinations. IoU and exact reconstruction are shown at 20 and 100 search steps. Lower values are better for conciseness.

| Agent | IoU | | Exact Reconstruction % | | Conciseness | # Parameters |
|-------|----------|-----------|----------|-----------|-------------|---------------|
| | 20 Steps | 100 Steps | 20 Steps | 100 Steps | | |
| real | 0.8644 | **0.9042** | 0.6232 | **0.6754** | 1.0168 | 3.02M |
| aug | **0.8707** | 0.8928 | **0.6452** | 0.6701 | **0.9706** | 3.02M |
| semi-syn | 0.8154 | 0.8473 | 0.5780 | 0.6104 | 1.0070 | 3.02M |
| syn | 0.6646 | 0.7211 | 0.4383 | 0.4835 | 1.0519 | 3.02M |

Table 3: Reconstruction results using random rollouts and gcn agents trained on human-designed data (real), a mixture of human-designed and semi-synthetic data (aug), semi-synthetic data (semi-syn), and synthetic data (syn).

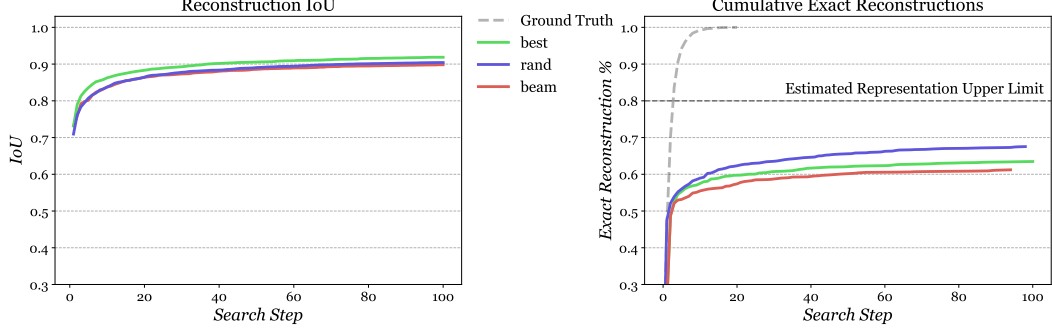

Figure 23: Reconstruction results over 100 search steps using the gcn agent with best first search (best), random rollout search (rand) and beam search (beam).

