# OpenReview forum: "Fusion 360 Gallery: A Dataset and Environment for Programmatic CAD Reconstruction"
_ICLR.cc/2021/Conference — Reject_

### Official Review · AnonReviewer3 · 2020-10-27
**Valuable new dataset, environment, and comprehensive evaluation metrics for 3D CAD design**

**Rating:** 7
**Confidence:** 4

**Review:**

Summary:

The paper presents the first large scale CAD construction sequence dataset together with environment that allows us to synthesize 3D CAD from these sequences. The dataset and environment are essential building blocks for applying machine learning algorithms to CAD design process. Furthermore, the paper proposes comprehensive evaluation metrics and a baseline method for predicting the sequence of 3D CAD design from a CAD model.

The dataset and environment are highly valuable to the research community. Although the proposed baseline and task have room for improvement, I believe this work merits publication.

Pros:
- The first large scale CAD dataset with the comprehensive construction sequence data.
- The environment can construct these sequence data and allow us to synthetically generate new data. Together with the 360 Fusion Gallery, this would be a great step towards learning human design processes.
- A good baseline method with comprehensive evaluation metrics.

Cons:
- The proposed task of CAD sequence reconstruction requires a target geometry in the same CAD format. This would not be called "raw geometry” as in the abstract. Raw geometry is often referred to as what we can derive from scan data including point clouds and meshes.
- As the aforementioned CAD reconstruction task is not very practical, adding another practical task such as reconstructing the sequence from raw geometry data is highly recommended.
- Regarding the evaluation metrics, it is not clear whether the conciseness is computed from only the successful reconstructions or the entire test set. If the former is the case, comparison using this metric across different approaches would not be fair. Otherwise, this does not necessarily provide a sense of how efficient the evaluated algorithm is. Clarification on this is recommended.

Questions/Remarks:
- In terms of accuracy, the augmentation rather hurts preciseness with more steps. Why is this happening? Is this due to the limited capacity of the proposed network? Discussion on this would be helpful.
- The following reference can be cited and discussed.
﻿"Modeling 3D Shapes by Reinforcement Learning” Cheng Lin, Tingxiang Fan, Wenping Wang, Matthias Nießner ECCV 2020

---

> ### Author Response · Authors · 2020-11-13
> **Response to Reviewer 3**
>
> Firstly, thank you for taking the time to review our submission. Below we provide additional discussion and outline planned changes based on your valuable feedback.
>
> **Re: Raw geometry wording**
> We agree this wording could be confusing and will update the paper to clarify.
>
> **Re: CAD reconstruction task practicality**
> We will update the paper to further motivate and clarify the practical application of CAD reconstruction from B-rep input. In summary, the CAD sequence, also known as the parametric modeling history, is often lost during data exchange, especially when models are exchanged between CAD applications using neutral file formats like STEP and IGES. The ability to edit the modeling history to modify, or entirely remove operations, is extremely valuable for users of CAD systems. The importance of this capability is reflected in the many industrial CAD packages that attempt to provide tools for restoring CAD history, such as [Inventor](https://knowledge.autodesk.com/support/inventor/troubleshooting/caas/sfdcarticles/sfdcarticles/How-to-recognize-features-in-Inventor-after-import-data-sets.html), [SolidWorks](http://help.solidworks.com/2019/english/SolidWorks/fworks/t_Recognizing_Features_Using_step_by_step.htm), [Creo](http://support.ptc.com/help/creo/creo_pma/usascii/index.html#page/part_modeling/feature_recognition_tool/About_Feature_Recognition.html), and [CATIA](https://www.cadcam-group.eu/catia-v5-part-design-features-recognition).
>
> These traditional algorithms operate by removing small features (typically holes or pockets) and re-applying them parametrically. Usually this strategy can restore the history for the later modeling operations, but can fail to completely rebuild the parametric modeling history from the first step. The task discussed in this paper restores the entire parametric modeling history, from the very first extrusion. Solving this task successfully has the potential to deliver benefit to CAD users beyond what is currently possible with traditional algorithms.
>
> **Re: CAD reconstruction from point clouds**
> The task of generating a CAD model with full parametric modeling history from a point cloud or triangle mesh is extremely valuable.  It is also very challenging technically. Very recent work has begun to tackle the first part of this problem by generating parametric surfaces and curves from point clouds [1, 2]. Although these papers show the promise of learning based approaches, they are not yet capable of producing geometry with the fidelity required to generate a solid model. Such a solid model could serve as input for CAD sequence reconstruction; the second part of the problem tackled in this work. A large amount of research is required to reach this goal and we hope the availability of the Fusion 360 Gallery reconstruction dataset will accelerate progress on this much more difficult task.
>
> 1. ParSeNet: A Parametric Surface Fitting Network for 3D Point Clouds, https://arxiv.org/abs/2003.12181
> 2. PIE-NET: Parametric Inference of Point Cloud Edges, https://papers.nips.cc/paper/2020/hash/e94550c93cd70fe748e6982b3439ad3b-Abstract.html
>
>
> **Re: Conciseness metric**
> The conciseness results in the paper are calculated from the exact reconstructions only. We consider conciseness to be a metric that describes the quality of a correct design. For practical use in a CAD tool, we consider an exact reconstruction to be a requirement. In general having a concise sequence is beneficial; designers will often revisit their designs to manually remove and consolidate lengthy sequences. The conciseness metric is designed to penalize reconstruction approaches that achieve good exact reconstruction performance, at the expense of less concise (i.e. longer) sequences.
>
> Currently we calculate conciseness as the geometric mean across all exact reconstructions for each condition (mpn, mlp etc…) with the number of exact reconstructions ranging from 928-1165 out of 1725 test files. Another approach is to hold the number of exact reconstructions constant across conditions by discarding designs that were not exactly reconstructed by all conditions. However we find that this ends up discarding longer sequences that are difficult to reconstruct and can improve the conciseness score in a potentially misleading manner. We welcome further discussion on the best way to quantify the conciseness of a construction sequence.
>
>
> **Re: Effect of augmentation on IoU**
> Yes, we agree this is an important point to better understand. We will provide additional results that compare the performance of neurally-guided search trained on synthetic, semi-synthetic (using human-designed sketches), and human design data. We believe these results will give us insight into the distribution of each dataset and provide evidence for the effect of data augmentation.
>
> **Re: Modeling 3D Shapes by Reinforcement Learning**
> Yes, this is an excellent reference to add. Thank you.

---

### Official Review · AnonReviewer2 · 2020-10-28
**Comprehensive submission, albeit somewhat domain specific to CAD community**

**Rating:** 5
**Confidence:** 3

**Review:**

The paper describes a new dataset of 3D geometry construction sequences based on sequential sketching (i.e. sets of two-dimensional curves) and extruding (i.e. axes, angles and distances for extrusion profiles from sketches) combined with Boolean solid geometry operations. The dataset comprises the resulting objects as well as the human designed sketch-extrude sequence that lead to the object geometry. The 8625 objects are available in three different geometry representations (boundary representation, mesh, and construction sequence in structured format). Furthermore the submission describes a new infrastructure ("gym") to train and evaluate algorithms that estimate such construction sequences, as well as evaluation metrics to gauge the efficiency and effectiveness of such estimation algorithms. The paper also provides a reference method for sequence estimation based on imitation learning as well as several baselines and their empirical evaluation with respect to the posed benchmarks.

### Strengths
**[S1]** The paper is written well and takes a comprehensive look on a new dataset from the perspective of the data itself, semi-synthetic generation of new data, evaluation metrics and benchmarking on the data as well as a reference method and its performance analysis.

**[S2]** I particularly appreciate the synthetic data generation/augmentation capability. This may enable more structured studies in the future, for instance with respect to specifics of the constructed geometry: Does complexity of the design geometry matter, does the topology make a difference in achievable performance?

**[S3]** New datasets are often valuable contributions to the community, in particular when they allow the discovery of new insight (however see W1 below)

### Weaknesses
**[W1]** The main contribution of the paper seems to me the database itself and the associated gym. While new datasets are a good contribution in general, I am not convinced that the target audience of ICLR will benefit in a substantial matter from the exposure. Specifically, I feel that the benefit of human-designed vs. synthetically generated designs is not worked out convincingly. What can I learn, what insight can I gain from the human designs that I cannot from, e.g., purely synthetical, procedural designs? Are there difference in the statistics? It would strengthen the paper to highlight evidence that the data does have benefit in the sense that I can achieve something substantially new.

**[W2]** The submission claims a "novel, neurally guided method", but I do not see the novelty laid out clearly: outside of "using common sketch and extrude CAD modeling operations from real human designs" in section 2, the novelty is vague and the relation to the state of the art imprecise. Is it the use of sketch and extrude operations in an otherwise previously exposed sequence learning task? Is it using human-designed sequences (then, see W1)? It would strengthen the case for the paper if the novelty would be exposed more concisely.

### Further comments:
I do appreciate the comprehensiveness across data, benchmarks, reference solutions and evaluation of the paper and such data and gyms can be an important piece of the machine and representation learning puzzle. As it is, I am concerned that the submission is not appealing to a sufficiently large audience at ICLR.

---

> ### Author Response · Authors · 2020-11-12
> **Response to Reviewer 2**
>
> Thank you for taking the time to review our submission. We outline below the planned changes in response to your helpful feedback.
>
> **Re: Value of human designed datasets**
> We agree it is important to make clear the value of learning from human data. We will provide additional results that compare the performance of neurally-guided search trained on synthetic, semi-synthetic (using human-designed sketches), and human design data. We agree that research on producing procedural designs is an important area that can also benefit from access to a human-designed baseline dataset.
>
> **Re: Novelty**
> We will work to clarify the novelty of our method in the paper. In summary, the face extrusion action representation (Section 4.1, Figure 4) is novel and enables data-driven CAD reconstruction on real world designs for the first time. By contrast, prior work focuses on CSG operations on geometric primitives that are rarely used in professional settings. By providing a dataset and new action representation we hope to encourage future work closely aligned to real world scenarios.
>
> **Re: Relevance to the ICLR community**
> We believe our work has high relevance to ICLR and the broader machine learning community. Specifically, our work aligns with the emerging area using program synthesis as a representation for graphical output such as 3D shapes [1,2], scenes [3] and sketches [4]. The Fusion 360 Gallery dataset provides such “CAD programs” complete with ground truth human designed program sequence. We believe new approaches to neurosymbolic reasoning can be developed and applied in this domain, following the lead of works such as [5,6].
>
> More generally, 3D shape datasets have been shown to provide significant value to the machine learning community at large. For example, ShapeNet [7] is an established 3D shape dataset that is widely cited across the machine learning, computer vision, and computer graphics communities. We believe CAD programs are a natural representation of human-designed shapes, and providing a dataset with this representation will be invaluable to the machine learning community.
>
> 1. Learning to Infer and Execute 3D Shape Programs, ICLR 2019, https://openreview.net/forum?id=rylNH20qFQ
> 2. UCSG-Net -- Unsupervised Discovering of Constructive Solid Geometry Tree, NeurIPS 2020, https://arxiv.org/abs/2006.09102v3
> 3. Learning to Describe Scenes with Programs, ICLR 2019, https://openreview.net/forum?id=SyNPk2R9K7
> 4. Learning to Infer Graphics Programs from Hand-Drawn Images, NeurIPS 2018, https://arxiv.org/abs/1707.09627
> 5. Learning Neurosymbolic Generative Models via Program Synthesis, ICML 2019, https://arxiv.org/abs/1901.08565
> 6. Multi-Plane Program Induction with 3D Box Priors, NeurIPS 2020, http://bpi.csail.mit.edu
> 7. ShapeNet: An Information-Rich 3D Model Repository, 2015 https://arxiv.org/abs/1512.03012

---

### Official Review · AnonReviewer4 · 2020-10-30
**Very interesting paper, unfortunately out of my field of expertise**

**Rating:** 8
**Confidence:** 1

**Review:**

I have to admit straight away that this paper is far from my field of expertise (computer vision, generative networks). I have not worked with CAD models, and I am not in expert in reinforcement/imitation learning. My review is thus written from the "educated outsider" viewpoint.

From that viewpoint, the paper is very strong. It introduces a meaningful problem (CAD modeling sequence reconstruction), motivates the need for a new task and the direction of research, describes the full task and the compromises it makes (face extrudes rather than sketch extrudes are considered in the simplified task). The paper then introduces a new training/test dataset and an environment for training agents, and evaluates a reasonable set of agents rather extensively.

As a slight criticism, I found that too many details are moved from Section 5 to the supmat. E.g. what is meant by MLP with
"an auto-regressive connection between the two target faces" was completely unclear before I looked into the supmat (and the phrase did not refer me to supmat either).
Otherwise, the paper is well-written and has nice visualizations (though they crash my PDF viewer) that aid understanding.

Overall, I really enjoyed reading the paper, and I am not able to identify any flaws. As far as I can judge based on my limited knowledge, the paper (together with associated dataset and environment) is likely to spur new research and to be impactful. I therefore give it a strong rating, but I cannot exclude that I have missed some flaws that would be identifiable by an expert.

---

> ### Author Response · Authors · 2020-11-13
> **Response to Reviewer 4**
>
> We appreciate you taking the time to review our submission. We also apologize for the issues with the PDF crashing your viewer. We will fix that when we post a revised version.
>
> **Re: Important details in the appendix**
> We will review Section 5 to ensure sufficient details are provided. Including a more detailed explanation of the auto-regressive connection. If there are any additional details you believe should be in the main paper, please let us know.

---

### Official Review · AnonReviewer1 · 2020-11-03
**Good dataset paper but insufficient evaluation of SOTA for the baseline approach**

**Rating:** 4
**Confidence:** 4

**Review:**

### Summary and Contributions
The CAD reconstruction problem is defined as the recovery of the sequence of modeling operations used to construct the CAD model, from the raw geometry input (triangle meshes, point clouds, or, B-reps). The paper proposes a novel dataset as well as a generic framework implementing an MDP (Markov Decision Process) formulation for training a neural CAD reconstruction agent.

This the first dataset which includes the sequence of CAD modeling operations as ground truth containing 8.5K+ human-designed real-world CAD models. The CAD reconstruction task is formulated as that of training a generic neural MDP agent and provided in an programming environment to the research community to train such models. Finally, a novel algorithm is provided as a baseline to solve the CAD reconstruction problem.

### Detailed Review
The following is the detailed review of the paper, organized into strengths and weaknesses subsections.

#### Strengths

##### Relevance and Significance
There is a large number of man-made objects that we interact with that are created using computer-aided design. The modeling steps, which are often lost, are important in understanding, editing, simulation and manufacturing such objects, and need to be reconstructed. This topic is of considerable importance to the CAD community. In addition, approaches, that can reverse engineer the generative process should be of general interest to the wider ML and computer vision community, as well.

##### Relation to Prior Art
The paper does a good job of presenting the prior art, identifying the challenges and need for the presented work. The dataset including the modeling sequences for real, human-designed CAD models is the first of its kind. Since it’s a novel problem (first dataset of the kind), the proposed baseline implementing a learning-based approach to this problem is novel as well (though simplistic, see below).

##### Reproducibility
Since the entire dataset and the baseline is released in an open-source programming environment, it should be easy to reproduce and verify the results.

##### Clarity
The paper is written well and is easy to understand.

#### Weaknesses

##### Methodology
This is largely a dataset paper. Introduction of a new dataset to the research community needs to demonstrate that the tasks to be solved on the dataset is not trivially addressed by the known state of the art. The paper falls short of demonstrating that.

1. Agent model: Two models are considered – MLP (trivial embeddings based on vertex features) and seemingly trivial embeddings of MPNs. The paper doesn't present enough details about the choices made for obtaining meaningful embeddings. This seems to be the heart of the approach and the authors don't present a compelling approach or a comparative evaluation of a set of choices based on the SOTA for graph embeddings, to model the agent.

2. Search: Similarly, the search is trivially implemented using a random rollout. There are much better search strategies available in the SOTA to bring to bear on the problem.

 In summary. I don't think that the presented baseline properly brings the SOTA to bear on the problem and thus demonstrates a need for additional research to be spurred on by this dataset.

##### Novelty
The presented approach is a straight-forward application of known techniques.

##### Empirical Evaluation
An important question to consider when proposing a presumably difficult new problem when addressed by known art is to investigate what aspect of the new problem really taxes the state of the art and to stage the experiments and analysis carefully for the same. There are many questions to be addressed here, for e.g. (a) How is stationarity (or, invariance over the topologies) achieved across training, validation and  test sets, (b) Does the training overfit (are the architectures used of enough capacity)?, (c) Is there generalization gap? If so, why? Etc.

### Assessment
Though the problem seems relevant and of significance to the research community, the dataset of considerable value, the paper doesn't make a strong case whether and how this problem challenges the known state of the art. In its current form, I do not recommend the publication of this paper.

---

> ### Author Response · Authors · 2020-11-11
> **Request for confirmation of SOTA baselines**
>
> Reviewer 1,
>
> Thank you for the thoughtful review. Allow us to first clarify the SOTA baselines.
>
> *Re: 1. Agent model*
> We look forward to running additional baselines to demonstrate that the CAD reconstruction task from B-rep input is not trivially addressed by known SOTA. Currently we plan to present further results using a Graph Attention Network (GAT) [1] and a Graph Isomorphism Network (GIN) [2]. Please advise if these two baselines are suitable for representing SOTA.
>
> *Re: 2 Search*
> In the appendix of the paper we currently provide results for two other search strategies: Beam Search and Best First Search. We will add additional plots and discussion to make the comparison of search strategies clearer and provide an update for feedback at that time.
>
> [1] https://arxiv.org/abs/1710.10903
> [2] https://arxiv.org/abs/1810.00826

---

### Author Response · Authors · 2020-11-19
**Revised Paper and Summary of Changes**

Dear Reviewers and AC,

Thank you again for your constructive comments. We have revised our paper accordingly and uploaded the latest version. The main changes include:

- We have added additional baselines using a Graph Isomorphism Network (GIN) and Graph Attention Network (GAT) to demonstrate that the CAD reconstruction task cannot be solved trivially.
- We have added an additional experiment comparing human-designed data to semi-synthetic and synthetic data. The results show the significant advantage of training with human-designed data.
- We have clarified a number of details in the main paper including: the novel aspect of our method, added details on the task motivation/practicality, clarified how conciseness is calculated, added discussion on the effect of data augmentation, clarified network implementation details, and added suggested references.
- We provide further details in the appendix including: a full table of results with combinations of agents (gcn, gat, gin, rand) with search (rand, best, beam), an additional plot comparing search strategies, a table of results comparing the performance of synthetic data, additional evaluation details, and a plot illustrating average reconstruction time.

Please don’t hesitate to respond with any additional feedback. Thank you.

---

### Decision · Program_Chairs · 2021-01-07
**Final Decision**

**Decision:**

Reject

**Comment:**

This paper received 4 reviews with mixed initial ratings: 4, 8, 5, 7. The main concerns of R1 and R2, who gave unfavorable scores, included limited methodological novelty beyond the data generation and insufficient empirical evaluation of state-of-the-art methods on the proposed dataset. The authors submitted a new revision with a summary of changes and provided detailed responses to each of the reviews separately: it addressed some of the concerns, but did not change the overall position of the reviewers.
AC agrees with R3 and R4 that the proposed dataset and the environment may have certain practical impact and enable new research in learning CAD reconstruction. However, the contributions are indeed specific to a narrow CAD community, and R1 felt that the paper needs another round of peer reviews before acceptance, as a significant number of new results have been added during the discussion stage. After discussion with PCs, the final recommendation is to reject.